# Reactive focal drug administration associated with decreased malaria transmission in an elimination setting: Serological evidence from the cluster-randomized CoRE study

Daniel J. Bridges[1]*, John M. Miller[1], Victor Chalwe[2], Hawela Moonga[3], Busiku Hamainza[3], Richard W. Steketee[4], Brenda Mambwe[1], Conceptor Mulube[1], Lindsey Wu[5], Kevin K. A. Tetteh[5], Chris Drakeley[5], Sandra Chishimba[1], Mulenga Mwenda[1], Kafula Silumbe[1], David A. Larsen[6]

1 PATH-Malaria Control and Elimination Partnership in Africa (MACEPA), National Malaria Elimination Centre, Chainama Hospital College Grounds, Lusaka, Zambia, 2 National Health Research Authority, Paediatric Centre of Excellence, University Teaching Hospital, Lusaka, Zambia, 3 National Malaria Elimination Centre, Zambia Ministry of Health, Chainama Hospital, Lusaka, Zambia, 4 PATH-Malaria Control and Elimination Partnership in Africa (MACEPA), Seattle, Washington, United States of America, 5 London School of Hygiene and Tropical Medicine, London, United Kingdom, 6 Syracuse University Department of Public Health, Syracuse, New York, United States of America

* dbridges@path.org

**Data Availability Statement:** Data deposited in OSF at following link: https://osf.io/hjf97/.

## Abstract

Efforts to eliminate malaria transmission need evidence-based strategies. However, accurately assessing end-game malaria elimination strategies is challenging due to the low level of transmission and the rarity of infections. We hypothesised that presumptively treating individuals during reactive case detection (RCD) would reduce transmission and that serology would more sensitively detect this change over standard approaches. We conducted a cluster randomised control trial (NCT02654912) of presumptive reactive focal drug administration (RFDA–intervention) compared to the standard of care, reactive focal test and treat (RFTAT—control) in Southern Province, Zambia—an area of low seasonal transmission (overall incidence of ~3 per 1,000). We measured routine malaria incidence from health facilities as well as PCR parasite prevalence / antimalarial sero-prevalence in an endline cross-sectional population survey. No significant difference was identified from routine incidence data and endline prevalence by polymerase chain reaction (PCR) had insufficient numbers of malaria infections (i.e., 16 infections among 6,276 children) to assess the intervention. Comparing long-term serological markers, we found a 19% (95% CI = 4–32%) reduction in seropositivity for the RFDA intervention using a difference in differences approach incorporating serological positivity and age. We also found a 37% (95% CI = 2–59%) reduction in seropositivity to short-term serological markers in a post-only comparison. These serological analyses provide compelling evidence that RFDA both has an impact on malaria transmission and is an appropriate end-game malaria elimination strategy. Furthermore, serology provides a more sensitive approach to measure changes in transmission that other approaches miss, particularly in very low transmission settings.

**Funding:** This work was supported by a grant from the Bill & Melinda Gates Foundation (OPP1134518 / INV-009984). The funder had no role in study design, data collection and analysis, decision to publish, or preparation of the manuscript.

**Competing interests:** The authors declare that they have no competing interests.

**Trial Registration:** Registered at www.clinicaltrials.gov (NCT02654912, 13/1/2016).

## Introduction

Malaria transmission continues to affect much of the world's population, despite renewed efforts for its elimination. While insecticide treated mosquito nets and artemisinin combination treatments have greatly reduced mortality from malaria, transmission persists and threatens resurgence. End-game elimination strategies are desperately needed. One such end-game malaria elimination strategy is Reactive Case Detection (RCD)–a type of contact tracing for malaria cases triggered by the detection of a confirmed case of malaria. RCD is widely deployed in Zambia as a Reactive Focal Test and Treat (RFTAT) response, whereby the index case's family members and neighbours within a 140m radius are tested and positive individuals treated [1]. RCD assumes that testing positive indicates peri-domestic malaria transmission. We hypothesised that RCD effectiveness could be improved by switching from RFTAT to Reactive Focal Drug Administration (RFDA) for the following reasons. First, RFTAT relies on a diagnostic that misses some low-density infections. In low transmission areas, infections tend to have lower parasite densities below the diagnostic limit of detection, therefore a significant proportion of the infectious reservoir could be missed [2, 3]. In contrast, RFDA treats all individuals present during a response. Second, successfully treating the human parasite reservoir does not affect the existing vector parasite reservoir. If locally infected vectors persist, reinfections will occur and the chain of transmission continue. Therefore, providing long-lasting chemoprophylactic protection through RFDA to the at-risk population will further reduce transmission. For example, dihydroartemisinin-piperaquine (DHAP) [4] has a markedly longer half-life of ~1 month, when compared to artmether-lumefantrine (AL) at ~7 days, the standard of care in Zambia [5]. Considering the low and focal nature of transmission and the need for inexpensive, community-implementable solutions, the more expansive population-wide deployment of Mass Drug Administration (MDA) would likely be overkill and could select for drug resistance [6].

Assessing intervention impact in pre-elimination settings is challenging. The standard endpoints of malaria intervention trials include; parasite prevalence; active infection incidence from cross sectional or cohort studies and passive incidence as measured from health facilities. When transmission nears elimination and is sporadic and low, sample sizes required to obtain a certainty of difference between interventions based upon parasite prevalence are impractically large. Active incidence is potentially useful but requires a study team to conduct repeated follow-up visits to ensure unbiased measures. Health facility incidence is attractive, as it is low cost and will have sufficient power, however it is prone to various biases including variations in care seeking behaviour and travel.

The presence of serological markers, i.e., antibodies, specific to *Plasmodium falciparum*, represents a powerful tool for assessing historical exposure. Historically, a limited set of markers were used to confirm national malaria elimination e.g. in Greece [7], but serology has not been broadly used as a tool to assess progress towards elimination. Recent technical advances have dramatically increased the number of antigens that can be simultaneously assessed i.e. multiplexed [8], allowing a range of targets associated with varying kinetics to be integrated and provide a fuller understanding of exposure. These approaches are now being applied to impact evaluations [9, 10]. Considering the need to assess elimination strategies in low transmission settings, serology represents an approach to increase the signal to noise ratio and

observe historical exposure trends through an attainable sample size. Herein we present the use of serological markers as a primary endpoint to assess progress towards malaria elimination, and more specifically the difference between two RCD approaches in Southern Province, Zambia, in the community-led responses for elimination (CoRE) study.

## Methods

### Intervention and participants

We conducted an unblinded cluster-randomized controlled trial to compare the impact of RFDA with DHAP against the standard of care—RFTAT with AL—on seroprevalence and malaria incidence in an area of seasonal low malaria transmission approaching elimination in Southern Province, Zambia. *Anopheles arabiensis is* considered to be the primary vector in the study area [11], although numerous other vectors do contribute to transmission [12]. Briefly, all health centers in Southern Province, Zambia maintain a cadre of volunteer community health workers (CHW) that provide RDT testing and treatment, with AL, for suspected malaria cases, and conduct RFTAT on confirmed cases living within a 140m radius of the index case [1]. Sixteen (16) health facility catchment areas (HFCA) from four districts were enrolled and randomised to receive either an RFTAT (standard of care) or RFDA (intervention) RCD response to an incident malaria case (Fig 1). The intervention began in May 2016 and was conducted for two years through May 2018.

In the control (RFTAT) arm, CHWs travelled to the home of incident malaria cases and tested all verbally consenting individuals within a 140m radius with a Malaria Ag P.f (Standard Diagnostics, Rep of Korea) rapid diagnostic test (RDT) and treated positive individuals with AL.

In the intervention (RFDA) arm, CHWs treated all individuals living with 140m from the incident case regardless of symptoms and without a diagnostic test. Children younger than 3 months old and pregnant women in the first trimester were excluded from the intervention and were offered a malaria RDT and AL treatment if positive (the standard of care). Written (adults), verbal (children aged 6–17 years old) and parental / guardian (children under 6 years old) witnessed consent was obtained from all eligible individuals who were then given a treatment dose of 4 mg/kg/day dihydroartemisinin and 18 mg/kg/day piperaquine (DHAP) for three days. In both arms, the first and last doses were directly observed by the CHW, or where the final dose had been taken, blister packs were checked. During the day 3 visit, adverse events were also recorded by the CHW.

Ethical approval was obtained from Western Institutional Review Board (1155095), the University of Zambia (011-10-14), the Zambia Medicines Regulatory Authority (CT 052), and the trial was registered at www.clinicaltrials.gov (NCT02654912, 13/1/2016).

### Evaluation study design

The RFDA intervention was evaluated using the primary outcome of seropositivity from an endline random survey of households. The antibody response to long-term antigens were assessed using a difference in difference approach while a post-only comparison was used for short-term antigens. The primary outcome deviated from the original protocol [13] in one respect -the maximum age of participants in the endline household survey was extended from 5 to 15 years old to enable a comparison to increase the sample size and allow comparisons to be made between the under 5 and over 5 age groups.

The evaluation also used two secondary outcomes of confirmed health facility malaria incidence from the Health Management Information System in Zambia, and 30- and 90- day RCD follow-ups from both arms. For the latter, a subset of CHW responses in both arms were

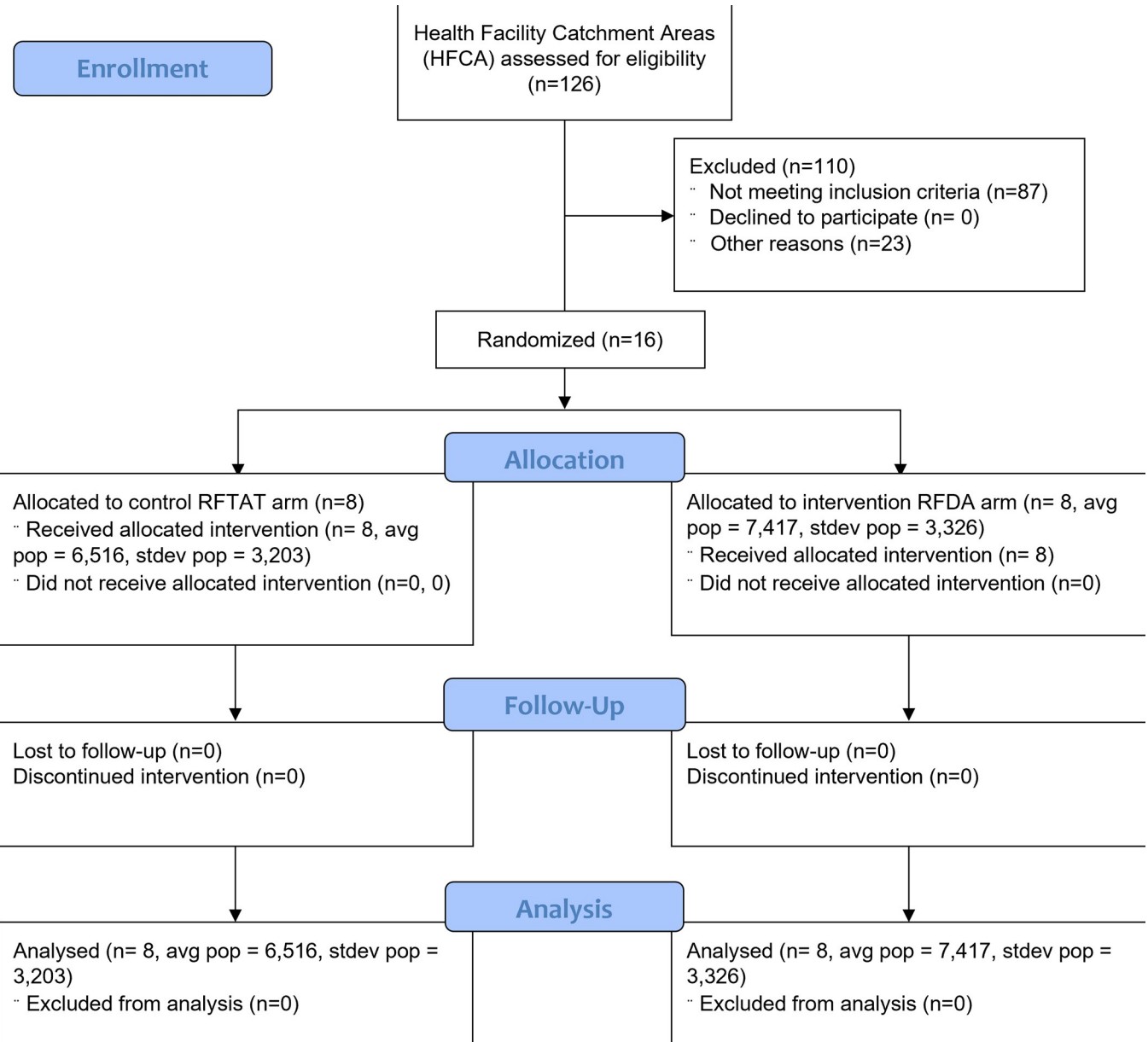

**Fig 1. Study participant flow.** Sixteen clusters were selected from all eligible health facility catchment areas in Southern Province, Zambia.

selected from each cluster via convenience sampling. In these responses, CHWs were accompanied by a research team with additional visits performed on days 30 and 90 to collect DBS for PCR testing.

## Data sources

A cross-sectional survey of randomly selected households in the 16 HFCA was performed at the end of the trial (May 2018). Households within the intervention area were randomly selected from satellite household enumerations, and field workers invited heads of household to participate in the survey. Participants received a standard household questionnaire (the 2015 Malaria Indicator Survey developed by the Zambian National Malaria Elimination

Centre [14]). Written consent was obtained for each parent/guardian, and dried blood spots (DBS) were collected on Whatmann Filter paper 3 for each child aged 1–15. A malaria RDT was also performed with RDT-confirmed infections treated according to national policy.

Weekly HFCA data, including clinical and laboratory confirmed malaria cases, total outpatients, RDT stock levels and monthly CHW data including RDT-confirmed cases and treatments dispensed was accessed through the national DHIS2 instance.

Environmental data was collected including; normalized digital vegetation index (NDVI) from the Landsat Tier 1 8-day NDVI collection aggregated to month (median); precipitation from the Climate Hazards Group InfraRed Precipitation with Station data collection [15] aggregated to month (sum); yearly night-time lights from the Visible Infrared Imaging Radiometer Suite [16]; and the digital elevation model (ASTER projection) from Google Earth Engine [17]. Environmental values were linked to each household using the Raster package [18] in R version 3.5.1 [19]. In cases of missing NDVI data due to high cloud cover, we used linear interpolation between nearest time points to impute data.

## Laboratory assays

**Serology.** Antibodies were eluted from a 3mm DBS punch (~2 μl of whole blood) and antibody titres for a range of *Plasmodium falciparum* antigens (**S1 Table**), determined using a Luminex based multiplex bead assay as described previously [20]. Controls, consisting of a six point and two-point serial dilution series of CP3 hyperimmune serum and WHO reference standard 10/198 [21] were run on each plate (**S2 Fig**). Only data with ≥ 30 beads / analyte /well, were included. 160 samples were completely excluded, while sample responses to eight antigens for 240 samples were excluded due to problematic standards [22]. Samples from infants (<1 year old) were not included.

**PCR.** DNA was extracted from a single 6 mm DBS punch (~13 μl of whole blood) using the QIAamp DNA mini kit (QIAGEN, Hilden, Germany). RDT-negative samples with two or more DBS were extracted in pools of ten while RDT-positive / PCR-pool-positives or single-spot DBS were extracted individually and stored at -20˚C. Extracted parasite DNA was detected in duplicate by photo-induced electron transfer PCR targeting the 18s rRNA locus [23] on a Light Cycler 480 real-time PCR machine (Roche, USA) and scored positive with duplicate cycle threshold values of < 40. A limiting dilution series of 3D7 reference *P. falciparum* was assayed 3 times in duplicate to estimate parasitaemia (**S1 Fig**).

**Outcome analysis.** We conducted an intention-to-treat analysis wherein children living in a health center catchment were assigned to either intervention or control based upon the health center rather than any participation in RFTAT or RFDA intervention. Seropositivity for anti-*Plasmodium* IgG during the simple random survey of children 1–15 years old served as the primary outcome of the study. (Children < 1 year old were excluded to avoid issues with maternal antibodies). A secondary study outcome of monthly confirmed incident malaria cases identified at the health center or in the community was also considered. All analyses were conducted using Stata (version 15.1). Finally, reinfections were recorded for individuals followed longitudinally (on days 30 and 90).

**Serology.** Each individual was classified as IgG positive or negative for each antigen using Finite Mixture Models (FMM) [24] of log-transformed mean fluorescence intensity. FMM threshold values for positivity to each antigen was set using a conservative posterior probability of < 0.01. IgG responses to antigens were a priori defined as long- or short-term markers of exposure based upon previous data [25] and experience.

*Long-term antigens.* Long-term malaria exposure was assessed using three classical *P. falciparum* markers—AMA-1, MSP1-19, and GLURP-R2 [26], with individuals classed as having

historical exposure if seropositive for any of the three antigens. The impact of RFDA was then assessed using a difference-in-differences analysis with child age as a proxy for time and the interaction term of child's age (<5 years or 5–15 years of age) and intervention allocation. A log-binomial regression was performed, with HFCA as a random intercept, adjusting for household wealth quintile based on a principal components analysis of owned assets, head of household education level, open or closed eaves, whether the child slept under an insecticide treated mosquito net the previous night, and whether the house had electricity (Equation 1, **S1 File**).

*Short-term antigens*. Short-term malaria exposure was assessed with CSP (full-length, Gennova), GEXP18, MSP2_CH150, H103/MSP11, HSP40 Ag1, and Hyp2 [20, 25], with individuals classed as having been exposed, if seropositive for any of the six antigens. RFDA impact was assessed using the difference in positivity between intervention and control arms in a post-only comparison using a log-binomial regression approach (Equation 2, **S1 File**). Three sensitivity analyses were performed to assess the robustness of the above by 1) changing the age cut-off to <4 and <6 years of age, 2) sequentially removing each antigen, and 3) sequentially removing each HFCA from the data to assess if any were overly influential.

**Health facility malaria incidence.** A generalized linear model with the HFCA as a random intercept and a negative binomial link due to overdispersion was used to assess the association between the arms and confirmed malaria incidence. Two separate measures, consisting of health centre cases or health centre and community cases (excluding RCD positives), of confirmed malaria incidence were considered.

Outcomes were standardised to the DHIS2 estimated HFCA population. Modelling was conducted using a priori hypothesized factors influencing malaria incidence. Environmental measures of NDVI, precipitation, altitude, night time light, number of RDT diagnostics performed each month, previous month's confirmed malaria cases, and a Fourier term to account for seasonality [27] were also included. Finally, the effect of the intervention on malaria incidence was examined using an interrupted time series approach (Equation 3, **S1 File**).

**RCD follow-ups (30- and 90-day).** PCR prevalence on days 30 and 90 were compared between the two arms using a Fishers exact test.

## Results

Between May 2016 and May 2018, a total of 668 confirmed malaria cases led to 692 RCD responses (Table 1) with a >93% response rate. RFTAT arm CHWs performed ~25% more RCD responses and enrolled more individuals per household (5.2) than in the RFDA arm (3), although ~16 times more treatment courses were dispensed in the RFDA arm. Response frequency fell during the trial in line with incidence (Fig 2).

### Cross-sectional survey

A total of 6,276 children (3,125 RFTAT, 3,151 RFDA) from 2,095 households (5,040 visited) were sampled in the post-intervention survey during April and May 2018. 16 children (0.25%, 95% CI = 0.13–0.38%) were malaria positive by PCR (7 RFTAT, 9 RFDA).

**Table 1. Number of individuals enrolled and treated in the CoRE study by arm.**

| Arm | Confirmed malaria cases | Number RCD responses | Number Households enrolled in RCD | Number enrolled in RCD | AL courses dispensed during RCD | DHAP courses dispensed |
|---|---|---|---|---|---|---|
| **RFTAT** | 345 | 392 | 749 | 3,953 | 118 | 0 |
| **RFDA** | 323 | 302 | 618 | 1,865 | 90 | 1,775 |
| **Total** | **668** | **692** | **1,367** | **5,818** | **208** | **1,775** |

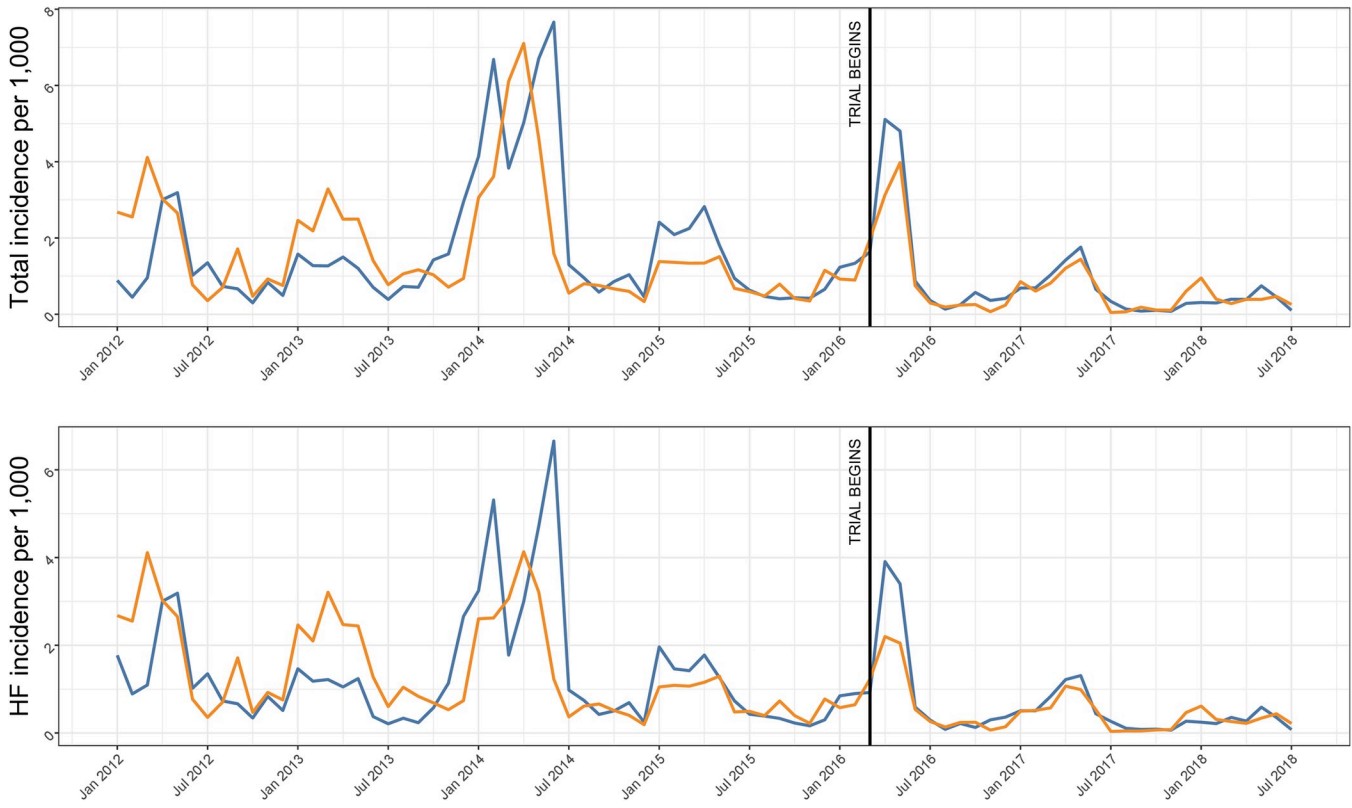

**Fig 2. Median confirmed population malaria incidence for the entire HFCA (total) or just those identified at the health facility (HF).** RFDA (blue) and RFTAT (orange) arms are shown.

**Long-term antigens.** Valid serological results for AMA-1, GLURP-R2, and MSP1-19 (Fig 3, S4 Fig) were available for a total of 5,152 children (2,554 RFTAT, 2,598 RFDA). Seropositivity for AMA-1 or MSP1-19 alone were ~12% while GLURP-R2 alone was 37% (**S1 Table**).

The log-binomial regression showed no difference in IgG seropositivity to long-term antigens between trial arms for children 5–14, while under 5's in the RFDA arm were 19% (95% CI = 4–32%) less likely to test seropositive than under 5s in the RFTAT arm (Table 2). Increasing or decreasing the age group cut-off by one year showed that children <6 years or <4 years in the RFDA arm were 18% and 14% less likely to test seropositive than those in the RFTAT arm respectively, although the latter was not statistically significant. Seropositivity increased proportionally with age (Fig 3) with under 5's 30% (95% CI = 21–37%) less likely to test seropositive compared to over 5's. A sensitivity analysis showed the magnitude of this finding was similar regardless of the antigen or HFCA combination used, although some combinations gave results that were not statistically significant (S2 Table). Wealth quintile, having electricity, having open eaves, and sleeping under an insecticide-treated mosquito net were not significantly associated with seropositivity to long-term antigens. The head of household having a secondary education or higher decreased the risk of testing positive (relative risk [RR] = 0.92, 95% CI = 0.85–0.99).

**Short-term antigens.** Valid serological results for CSP, GEXP18, H103/MSP11, HSP40 Ag1, Hyp2, and MSP2_CH150 were available for a total of 5,099 children, with 2,677 in the intervention arm and 2,422 in the control arm, with aggregate seropositivity of 2.6% (S3 Table). The risk of testing seropositive with a short-term malaria antigen increased steadily with age (one-year increase RR = 1.07, 95% CI = 1.02–1.11). Children in intervention areas

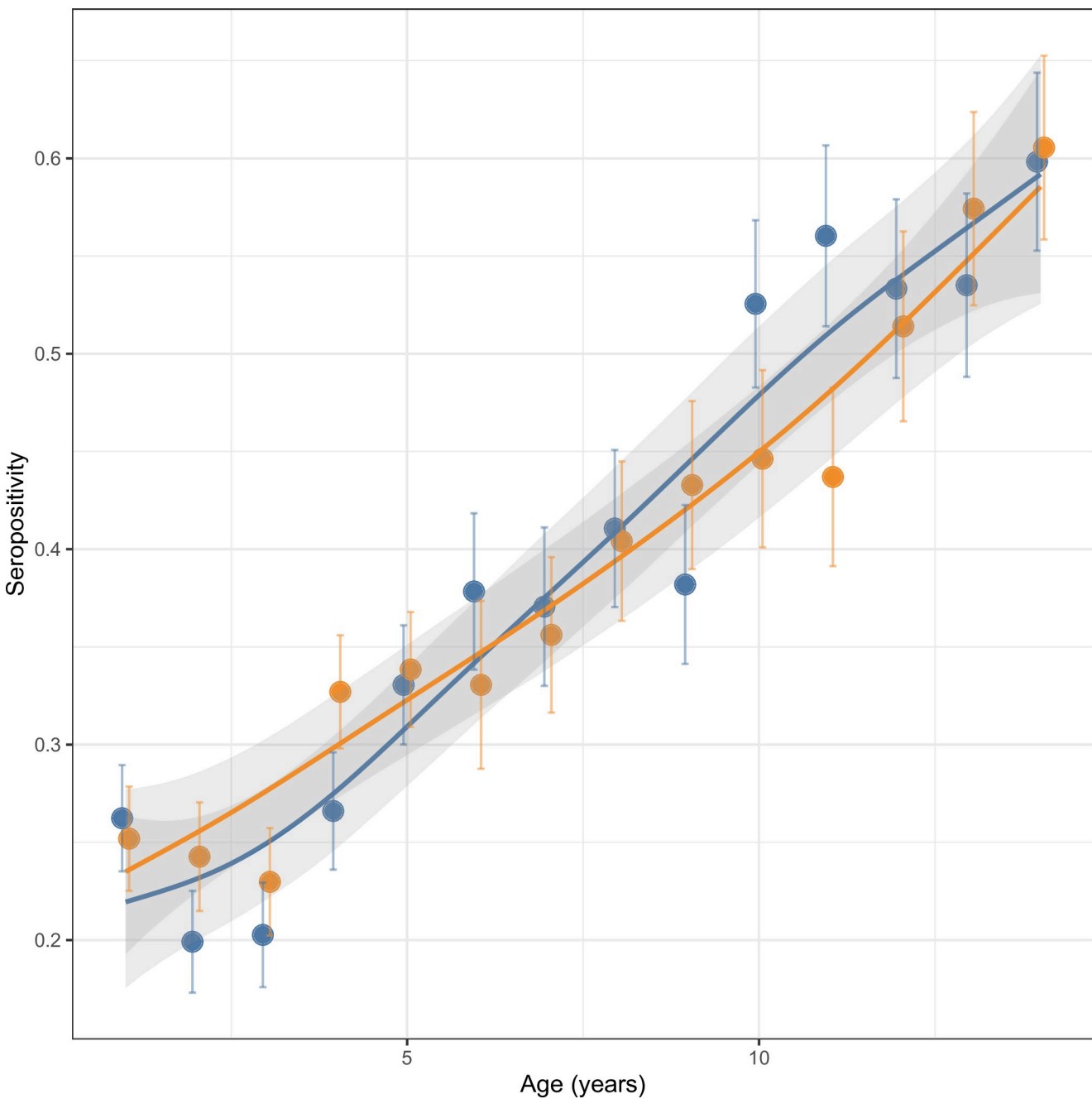

**Fig 3. Seropositivity by trial arm and age for any of AMA-1, GLURP-R2, or MSP1-19 antigens in a post-only simple random sample.** Data are fitted using a loess smoother function and 95% confidence intervals. RFTAT control (orange) and RFDA intervention (blue) arms are shown accordingly. Plots for individual catchments are shown in S3 Fig.

were 37% (95% CI = 2–59%) less likely to test seropositive with any of the short-term malaria antigens than children in control areas (Table 2, Fig 4). A sensitivity analysis showed the magnitude of this finding was similar regardless of the combination of antigens or HFCA used, although some combinations gave results that were not statistically significant (S2 Table).

Children living in households with electricity (RR = 0.45, 95% CI = 0.21–0.98) and households whose heads had higher education (RR = 0.72, 95% CI = 0.49–1.06) were less likely to

**Table 2. Results from a log-binomial regression model of seroprevalence to long-term antigens (AMA-1, GLURP-R2, MSP1-19, N = 5,152 unadjusted, 5,085 adjusted, individuals), and short-term antigens (GEXP18, H103/MSP11, HSP40 Ag1, Hyp2, CSP, and MSP2_CH150, N = 5,100 unadjusted, 5,036 adjusted, individuals) in 16 HFCA.**

| | Unadjusted RR (95% CI) | p-value | Adjusted RR (95% CI) | p-value |
|---|---|---|---|---|
| **Long-term antigens** | | | | |
| **RFTAT arm** | Ref. | Ref. | Ref. | Ref. |
| **RFDA arm** | 1.03 (0.85–1.24) | 0.748 | 0.99 (0.82–1.20) | 0.951 |
| **Age 5–14** | Ref. | Ref. | Ref. | Ref. |
| **Age under 5** | 0.70 (0.62–0.78) | < 0.001 | 0.70 (0.63–0.79) | < 0.001 |
| **Arm X age interaction** | 0.83 (0.70–0.99) | 0.037 | 0.81 (0.68–0.96) | 0.016 |
| **Short-term antigens** | | | | |
| **RFTAT arm** | Ref. | Ref. | Ref. | Ref. |
| **RFDA arm** | 0.64 (0.40–1.03) | 0.066 | 0.62 (0.39–0.97) | 0.038 |

Adjusted model included factors of wealth quintile, head of household education, household electricity access, whether house had open eaves or not, whether the child slept under an insecticide-treated mosquito net the previous night and age.

test positive for a short-lived malaria antigen. Wealth quintile, having open eaves, and sleeping under an insecticide-treated mosquito net were not associated with short-term antigen seropositivity.

**Health facility malaria incidence.** From 2012–2018, malaria incidence declined in the study area from ~3 to < 2 cases per 1,000, while 2014 and 2016 showed higher than average

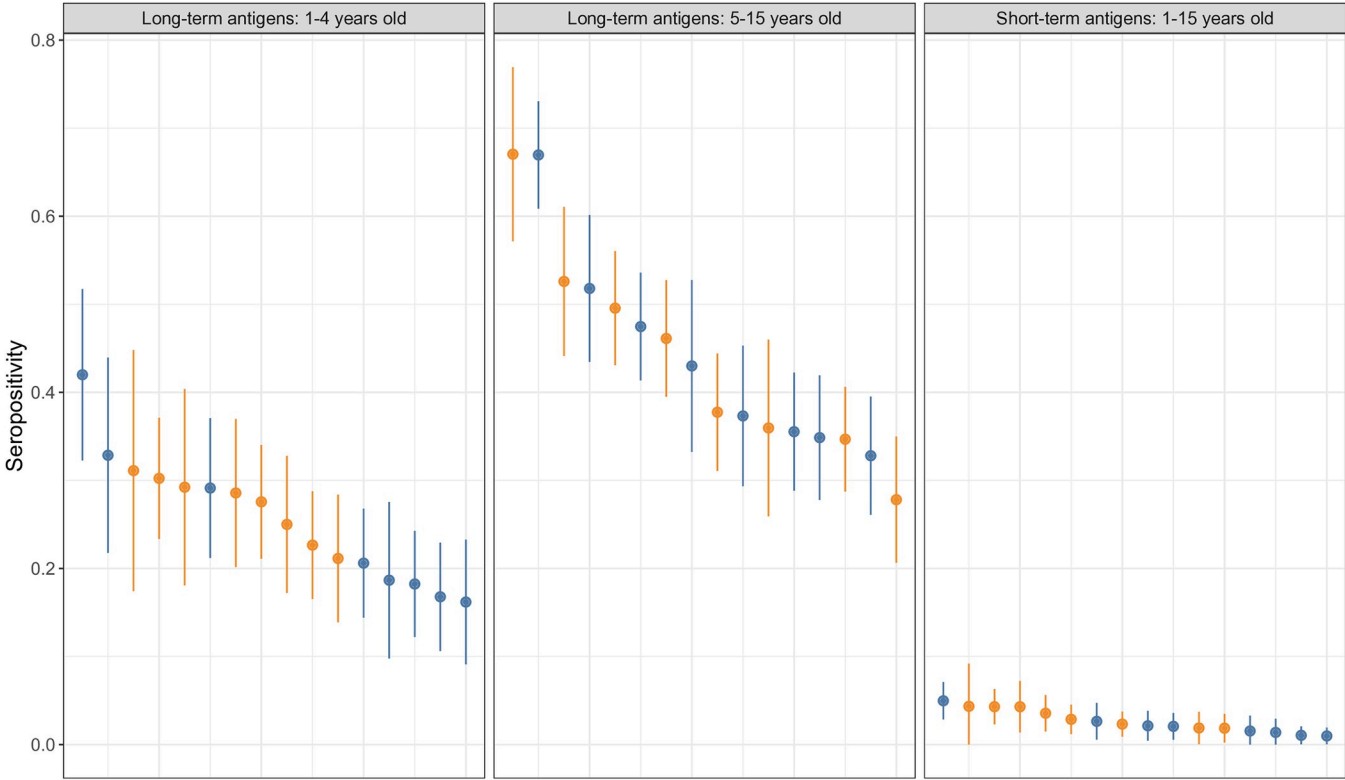

**Fig 4. Seropositivity for each of the health facility catchment populations to long-term antigens, stratified by age (left and middle), and short-term antigens (right).** Health facilities are ordered in each panel according to maximum seropositivity observed. RFTAT control (orange) and RFDA intervention (blue) arms are shown accordingly. Bars show 95% confidence intervals.

 

**Table 3. Results from interrupted time series analyses of confirmed malaria incidence.**

| | Unadjusted IRR (95% CI) | p-value | Adjusted IRR (95% CI) | p-value |
|---|---|---|---|---|
| **Confirmed malaria incidence at health centers and by community health workers** | | | | |
| RFTAT arm | Ref. | Ref. | Ref. | Ref. |
| RFDA arm | 1.01 (0.84–1.21) | 0.948 | 1.19 (0.97–1.47) | 0.097 |
| Time (month as continuous since Jan 2012) | 0.99 (0.98–0.99) | < 0.001 | 0.99 (0.98–0.99) | < 0.001 |
| Intervention time (month as continuous since Mar 2016 in RFDA arm only) | 1.00 (0.99–1.02) | 0.811 | 1.00 (0.99–1.01) | 0.985 |
| **Confirmed malaria incidence at health centers only** | | | | |
| RFTAT arm | Ref. | Ref. | Ref. | Ref. |
| RFDA arm | 1.07 (0.87–1.31) | 0.533 | 0.96 (0.75–1.22) | 0.713 |
| Time (month as continuous since Jan 2012) | 0.98 (0.97–0.98) | < 0.001 | 0.99 (0.98–0.99) | < 0.001 |
| Intervention time (month as continuous since Mar 2016 in RFDA arm only) | 1.01 (0.99–1.02) | 0.406 | 1.00 (0.99–1.02) | 0.811 |

N = 1,264 facility-months, 16 health facilities. Models adjusted for seasonality using a sinusoidal function. Adjusted analysis also included the following factors: lagged confirmed malaria cases (1 month), precipitation, NDVI, night-time light, and the number of malaria tests conducted that month.

malaria cases (Fig 2). After adjusting for environmental factors, the decline from 2012–2018 was a steady 15% (95% CI = 10–20%) reduction per year for all HFCA incident malaria cases. The intervention arm had 19% (95% CI = -4–47%) more confirmed HFCA cases during the trial period. Higher NDVI (more vegetation), higher precipitation, and lower night-time light were associated with higher malaria incidence as expected, although there was no difference between the two arms.

An interrupted time series analysis showed no difference in HFCA malaria cases (Fig 2) between the trial arms (Table 3) for total HFCA confirmed cases (IRR = 1.00, 95% CI = 0.99–1.01) or for health facility only confirmed cases (IRR = 1.00, 95% CI = 0.99–1.02).

### RCD follow-ups (30- and 90-day)

RCD responses followed up on days 30 and 90 identified a limited number of infections at these timepoints (S4 Table). A Fishers exact test was suggestive of the RFTAT control arm being inferior to the RFDA intervention arm at preventing reinfection by Day 30. However, by day 90 there was no difference between the two arms. In both arms, there was a high loss to follow-up of ~23% by day 30 and ~35% by day 90.

### Adverse events

A total of 123 people reported an adverse event (AE), all of which were recipients of DHAP in the RFDA arm. Of the symptoms reported, the majority were headache (20%), abdominal pain (17%), dizziness (17%) or nausea (16%). All AE were mild, self-resolving and did not require any clinical intervention. The number of reported AE were at or below the expected number for DHAP. No AE were reported for AL although the same AE data collection system was in place. This may be due to the familiarity of in Zambia where it has been the frontline treatment for malaria since 2003.

### Discussion

We performed a trial to assess the impact of RFDA (intervention) against RFTAT (control) RCD responses in an area of very low transmission. Both RFTAT and RFDA appeared to be well received by the community and no serious adverse events were reported.

## Serological marker analysis

Given the limitations in measurable outcomes for assessing malaria interventions using standard transmission metrics, we devised two statistical comparisons for evaluating the CoRE trial using serological markers in a post-only cross-sectional household survey. First, we leveraged long-lasting IgG responses (> 5 years) into a difference-in-differences (DID) [28] comparison, that examines the difference between pre- and post-intervention measures (the differences) between intervention and control groups (the difference). For the CoRE study this means comparing how differently the two arms of the trial changed over time. Typically, a DID approach requires a pre-intervention measure, but as we applied it here, we used seropositivity in children aged 5–15 as a measure of pre-intervention exposure. Children aged 5–15 could become seropositive during the trial rather than before, which is a limitation to this approach. To account for this limitation, secondly we performed a post-only comparison of short-term (longevity < 1 year) antigens. Taken together, the DID of long-lived antigens strengthens the claim of causal inference by accounting for pre-intervention differences between trial arms and the post-only comparison of short-lived antigens reduces the influence of the misattribution limitation, i.e., seroconverting to long-lived antigens after the trial began but at an older age. Our approach does not require a pre-intervention survey (half the cost), but the pathogen of interest does need to have both long- and short-lived serological markers identified, which for *P. falciparum* have been defined [25, 26]. As the level of false positives in the population is better understood, more accurate metrics of exposure, especially for short-term antigens, may be determined.

Most serological surveys calculate seroconversion rates (SCR), i.e., the modelled rate that antibodies are acquired, by performing all-age cross-sectional surveys, plotting the antibody titre against age and then fitting the data to a model [26]. Acquisition is initially linear, i.e., characterised almost exclusively by seroconversion, but levels off as seroreversion becomes significant. The early linear portion defines the SCR, with higher transmission intensity skewing the fit towards younger (higher SCR) or older (lower SCR) age groups. The CoRE study was not only performed in an area of low transmission intensity, but only those under 15 years old were enrolled, therefore fitting standard SCR models was not applicable.

## Study impact

Three alternative methods for measuring study impact were assessed. First, a simple comparison of HFCA malaria incidence was performed (Fig 2, Table 3). While this analysis did not identify any significant difference between the two arms of the study, it showed that incidence declined by ~40% during the trial. This decline was observed across Southern Province, Zambia and coincided with lower-than-average rainfall. Considering that transmission was low to start with and decreased further during the trial, it is not surprising that HFCA incidence was comparable between the two arms (Table 3). This demonstrates the difficulty of measuring intervention impact using traditional outcomes in a malaria elimination setting. Second, reinfections were assessed in a subset of RCD responses at days 30 and 90 (S4 Table). These data were suggestive of RFDA reducing infections on day 30, but that this effect had disappeared by day 90. Considering the high loss to follow-up, the small numbers involved and therefore the potential for sample bias to influence this result we found it supportive of RFDA being superior to RFTAT, but not conclusive.

The final approach taken was to perform an endline cross-sectional survey to look for current infections (PCR) and malaria exposure (serology). As expected, PCR identified a very limited number of infections (n = 16, 0.25%), and not enough to make a meaningful comparison between the two arms. At this low level of prevalence, sample sizes required to estimate a significant difference between arms increase enormously. It is possible that increasing the survey

sample could have generated enough data to compare parasite prevalence between the arms, however the sample size increase (and related cost) would have likely required a complete census rather than a survey.

In contrast, the multiplex serological assay provided a rich dataset for a range of antigens that together showed a wide population seropositivity range two orders of magnitude higher than PCR positivity (0–27%, S3 Table). Using different combinations of these antigens enabled the arms to be compared over more extended (long-term antigens) or more recent (short-term antigens) exposure history timescales. We found significant reductions in aggregate seropositivity in children under 5 in the RFDA arm to both long-term and short-term antigens of 19% (95% CI = 4–32%) and 37% (95% CI = 2–59%) respectively (Fig 3, Table 2). This strongly supports the hypothesis that RFDA reduces exposure to *P. falciparum*. Interestingly, this reduction was seen despite similar numbers of incident malaria cases recorded in both arms and demonstrates the limitations of routine data [6]. While we expected to see a difference in long-term antigens, considering that individual short-term antigen seropositivity was around 1% (S3 Table) combining the antigens enabled a significant result to be identified despite the rarity of the outcome. Overall, these data provide compelling evidence that RFDA both has an impact on malaria transmission and that it is more effective than RFTAT. Furthermore, RFDA is intrinsically quicker and easier to implement, requiring only treatments to be dispensed without testing. This could improve the number of responses performed as well as the timeliness of a response, both of which will likely further increase impact. To maximise population coverage, it is possible that treatments could be left for individuals absent during a response, although adherence and safety may be problematic. Alternatively, efforts could be made to expand the CHW network such that multiple repeat visits could be made more easily. While a more formal and in-depth cost and effectiveness comparison is needed, these promising features taken together with recent results from Namibia, that also showed RFDA to be superior to RFTAT [29], suggest that RFDA should be seriously considered to be implemented in low transmission settings and / or replace RFTAT approaches.

## Study limitations

While HFCAs were randomised to a study arm many fewer people were enrolled per household in the RFDA arm. This may reflect RFDA CHWs incorrectly excluding individuals, although no evidence for this was identified. While significant community sensitisation efforts were performed before and during the trial, we believe this discrepancy likely reflects a higher refusal rate in the RFDA arm. However, if true, this lower intervention exposure would bias the results toward the null and make finding a significant effect less likely.

The post-only comparison of seroprevalence is limited in that there was no pre-intervention seroprevalence estimate. We opted to exclude the pre-intervention seroprevalence estimate in order to ensure intervention fidelity, as ethically we would be required to treat every malaria infection found during a baseline survey. This would make the baseline survey itself a mass testing and treatment event, which does have an effect on malaria transmission [30]. We have attempted to account for the lack of a pre-intervention baseline by estimating seroprevalence by age group. The DID analysis used the *a priori* cutoff of 5 years. Increasing this to 6 years had no effect, but decreasing it to 4 years removed statistical significance. As age is related to the probability of having been infected, this result may simply highlight the low levels of transmission in the study area, whereby not enough infections have occurred in this smaller age group to reach significance. Previous studies of malaria elimination have used seroprevalence by age group as an indicator of transmission, but to our knowledge we are the first to use these measures in an intervention trial.

## Conclusions

In very low transmission settings, such as in this study, standard approaches to measuring transmission fail. We therefore used longitudinal follow-ups and serology to assess the impact of two RCD responses. While longitudinal follow-up data suggested that RFDA was more effective at reducing malaria prevalence than RFTAT, the effect was temporary and not conclusive. Serological analysis, however, clearly showed that the RFDA intervention reduced malaria transmission above and beyond the RFTAT (standard of care) approach. This adds to the body of evidence that in low transmission settings, serology is an appropriate method for assessing transmission. In summary, this work supports the implementation of RCD and specifically RFDA to reduce malaria transmission in very low transmission areas to push toward local malaria elimination.

## Supporting information

**S1 Checklist. CONSORT checklist.**
(DOC)

**S1 Fig. Reference standard curves for *P. falciparum* 3D7 strain showing the relationship between parasitaemia and Ct / DNA concentration.** The assay was performed in duplicate (n = 3).
(TIF)

**S2 Fig. Standard curves for dilution series of hyper-immune CP3 sera for representative long-term (AMA-1) and short-term (Hyp2) *P. falciparum* antigens across all plates.**
(TIF)

**S3 Fig. Seropositivity for each health facility by trial arm and age for any of AMA-1, GLURP-R2, or MSP1-19 antigens in a post-only simple random sample.** Data are fitted using a loess smoother function and 95% confidence intervals (grey shaded area). RFTAT control (black) and RFDA intervention (red) arms are shown accordingly.
(TIF)

**S4 Fig. Reverse cumulative plots for each *P. falciparum* antigen stratified by trial arm.** RFTAT control (orange) and RFDA intervention (blue) arms are shown accordingly.
(TIF)

**S1 Table. List of antigens used in the serology multiplex bead assay with positivity observed and numbers.**
(DOCX)

**S2 Table. Sensitivity analyses assessing the effect of removal of an antigen or a health facility on outcome.**
(DOCX)

**S3 Table. Population seroprevalence of a selection of long- and short-term malaria antigens.**
(DOCX)

**S4 Table. PCR results and loss-to follow-up for RCD follow-ups on days 30 and 90.AMA.**
(DOCX)

**S5 Table. Day 1 individuals recruited for RCD follow-ups on days 30 and 90 by age and gender.**
(DOCX)

**S6 Table. Individuals recruited to the post-only endline survey for serology by age and gender.**
(DOCX)

**S1 File. Log binomial regression equations used for analysis.**
(DOCX)

**S2 File. Ethics clearance.**
(PDF)

**S1 Protocol.**
(DOCX)

## Acknowledgments

The authors would like to thank all the study respondents, all the CHWs and to the Zambia Ministry of Health at all levels. We would like to thank the research team, Jenala Nyangu, Chama Chishya, Dumisani Munsaka, Abson Maleka, Barbara Syankwilimba, Innocent Muleta and Twaambo Simanga for their tireless efforts in the field along with data safety monitoring board members, Sebastian Hachizovu, Belden Hamuyube, Chongo Kenneth Chibwe, Christine Manyando, Benjamin Bellows, and Oscar Mwiinde who provided critical feedback throughout the study. We would like to thank Eric Rogier for supplying antigens and feedback on this manuscript. We would like to acknowledge William Moss and Anna Winters for useful discussions when conceptualising the study.

## Author Contributions

**Conceptualization:** Daniel J. Bridges, Richard W. Steketee, David A. Larsen.

**Data curation:** Daniel J. Bridges, David A. Larsen.

**Formal analysis:** Daniel J. Bridges, Chris Drakeley, David A. Larsen.

**Funding acquisition:** Daniel J. Bridges, John M. Miller, Richard W. Steketee.

**Investigation:** Daniel J. Bridges, John M. Miller, Brenda Mambwe, Conceptor Mulube, Sandra Chishimba, Mulenga Mwenda, Kafula Silumbe.

**Methodology:** Daniel J. Bridges, Brenda Mambwe, Conceptor Mulube, Lindsey Wu, Sandra Chishimba, Mulenga Mwenda, David A. Larsen.

**Project administration:** Daniel J. Bridges, John M. Miller, Kafula Silumbe, David A. Larsen.

**Resources:** Richard W. Steketee, Kevin K. A. Tetteh, Chris Drakeley.

**Software:** David A. Larsen.

**Supervision:** Daniel J. Bridges, Victor Chalwe, Hawela Moonga, Busiku Hamainza, Kafula Silumbe.

**Validation:** Kevin K. A. Tetteh, Chris Drakeley, David A. Larsen.

**Visualization:** Daniel J. Bridges, David A. Larsen.

**Writing – original draft:** Daniel J. Bridges, David A. Larsen.

**Writing – review & editing:** Daniel J. Bridges, John M. Miller, Victor Chalwe, Hawela Moonga, Busiku Hamainza, Richard W. Steketee, Brenda Mambwe, Conceptor Mulube,

Lindsey Wu, Kevin K. A. Tetteh, Chris Drakeley, Sandra Chishimba, Mulenga Mwenda, Kafula Silumbe, David A. Larsen.

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
