## [Decision Letter · Decision Letter 0]

16 Jun 2022

PGPH-D-21-00549

Reactive focal drug administration decreases malaria transmission in an elimination setting: serological evidence from the cluster-randomized CoRE study

Dear Dr. Bridges,

Thank you for submitting your manuscript to PLOS Global Public Health. After careful consideration, we feel that it has merit but does not fully meet PLOS Global Public Health’s publication criteria as it currently stands. Therefore, we invite you to submit a revised version of the manuscript that addresses the points raised during the review process.

The Reviewers raised several methodological and formatting concerns. Please revise the paper to address these concerns, being sure to clearly highlight any limitations and discuss the implications of these limitations for study findings.

We look forward to receiving your revised manuscript.

Kind regards,

Manisha A. Kulkarni

Academic Editor

Journal Requirements:

1. Please include additional information regarding the survey or questionnaire used in the study and ensure that you have provided sufficient details that others could replicate the analyses. For instance, if you developed the survey or questionnaire as part of this study and it is not under a copyright more restrictive than CC-BY, please include a copy, in both the original language and English, as Supporting Information. If the questionnaire is published, please provide a citation to the questionnaire.

2. In your Methods section, please provide additional information about the participant recruitment method and the demographic details of your participants. Please ensure you have provided sufficient details to replicate the analyses such as: 

a) a table of relevant demographic details, 

b) how participants were recruited.

3. Please provide additional details regarding participant consent. In the ethics statement in the Methods, please ensure that you have specified what type of consent you obtained (for instance, written or verbal, and if verbal, how it was documented and witnessed). If your study included minors, state whether you obtained consent from parents or guardians.

4. Please amend your detailed Financial Disclosure statement. This is published with the article. It must therefore be completed in full sentences and contain the exact wording you wish to be published.

State what role the funders took in the study. If the funders had no role in your study, please state: “The funders had no role in study design, data collection and analysis, decision to publish, or preparation of the manuscript.”

5. Please provide separate figure files in .tif or .eps format and ensure that all files are under our size limit of 10MB.

Reviewer #1: Thank you very much for sending me this paper for review. The investigators studied two interventions in southern Zambia:

a) Reactive presumptive treatment

b) Reactive test and treat

Both intervention are timely and relevant as national malaria control programmes aiming for malaria elimination.

The authors found at the end of the study period the percentage of seropositive was lower in presumptive treatment group (a) than in the test in treat group (b). This result makes intuitively sense as the authors explain. The diagnostic tools to detect very low density, subclinical infections in field settings are not yet available. Treating everybody within a certain radius around the index patients is logistically easier and likely to be more effective. The investigators did not find a difference in parasite prevalence or incidence which are the much more frequently used outcomes of similar, large programmatic malaria intervention studies. Serology is a complex endpoint as it includes a number of antibodies and requires sets of derivations to determine who is infected and who is not. When incidence and prevalence estimates suggest that there is no difference between the two interventions it is courageous to say that the rather derivative serology results indicate the true impact of the intervention and incidence and prevalence estimates are underpowered hence as the authors suggest.

There are two major design flaws which will make it difficult to convince readers:

There is no control group - which did not receive an intervention. We can therefore say that one intervention is as good or better than the other but we do not know the absolute impact on malaria. The fact that the malaria burden decreased over the study period suggests that the interventions made a positive impact but alternatively this decrease could be caused by other factors than the study. As the authors write "This decline was observed across Southern Province, Zambia and coincided with lower-than-average rainfall." Suggesting a secular event independent of the study interventions.

Second the investigators did not estimate the baseline serostatus in their study population. The authors may be right that the presumptive treatment reduced transmission more than the test and treat intervention did. Alternatively, the seroprevalence in the test and treat clusters was already higher at baseline. The authors provide indirect, unvalidated evidence that this is not the case as there was a difference in seroprevalence by age group. With all due respect I find this speculative and far from convincing. To make the serology data convincing the authors would have to show that the baseline seroprevalence was similar in both study arms.

It is probably impossible to address these flaws retrospectively. The only appropriate way to deal with this problem is to address these challenges clearly and honestly and then discuss the implications of these major limitations. The project must have been a major effort in terms of human and financial resources. The data are interesting and should be published but the data don't support the conclusion "These results provide robust confidence that the RFDA intervention reduced malaria transmission …" the seroprevalence estimates are flawed without a baseline estimate and the more direct, easy to interpret incidence and prevalence data suggest there is no difference between the treatment arms. The authors have to make this clear throughout the paper starting with the title.

In the discussion the authors say "PCR identified a very limited number of infections (n = 16, 0.25%), and not enough to make a meaningful comparison between the two arms..." suggesting that if only more case would have been detected the desired outcome would have been reached. This conclusion is entirely inappropriate. The authors wish that their work made an impact and if only their study had more power, they could have shown such an impact. While such an interpretation is entirely understandable there is the alternatively interpretation that their intervention made no impact. They can't disprove the null hypothesis and therefore have to accept the null hypothesis. The statement "It is possible that increasing the survey sample could have generated enough data to compare the arms, however the increase, and therefore cost, would have been dramatic and likely required a complete census rather than survey." Sorry, but you do not have the evidence to support this statement. It suggests that the authors are not familiar with the basic elements of hypothesis testing. We always hope that our interventions work but if there is no evidence, we have to accept that we failed to disprove the null hypothesis as disappointing and painful as it may be.

If the authors decide to revise their paper and address these major concerns they may wish also to address the following points:

The consort checklist for individually randomised trials is not very appropriate for cluster randomised trials. Please consider complying with the checklist for cluster randomised trials?

Figure 1, the consort chart illustrating the patient assembly is not very helpful. It would be much more interesting to see how many participants were recruited in each study arm and how many samples were analysed in the serology study. This is nicely explained in the consort checklist for cluster randomised trials.

At one point the authors refer to the "intervention arm". Since we are dealing with two arms each receiving different interventions this statement is not helpful.

The figures have no embedded legends it is difficult to guess what the colours mean. What is shown along the x-axis in Figure 4?

The imbalance of adverse events in the treatment arms is surprising. It would be interesting to know how the investigators explain this finding.

Overall, it does not help that the preparation of the manuscript is rather sloppy. Error warnings "Error! Reference source not found." appear repeatedly in the manuscript which should have been corrected before approving the submission.

Reviewer #2: Please indicate the major vector that was transmitting malaria. The flight range of this vector will determine the radius of the area of reactive case detection with reference to the index case. For example the flight range of An funestus is much smaller that that of An gambiae s.l.

It would be useful to indicate the half life of the short live verses the long lived antibodie in the

discussion. The authors should also discuss the rate of decay of the short-lived antibodies during chemotherapy with both drug treatment regiments.

As an alternative, would it have been feasible to use the rate of short lived antibody decay as a measure of diminishing transmission ? See Ototo 2011

Reviewer #3: This manuscript deals with a very important problem when striving to malaria elimination: how to reduce malaria and evaluate progress in areas with a low level of malaria. They present results from a trial that contrasts a reactive case detection strategy with a reactive case and presumptive treat scenario, with innovative tools to evaluate progress in both scenarios. The trial was entered in a registry before the start of the study, and protocol, and data have been made available to the reviewers, enhancing full transparency. However, the manuscript can be improved with a better description of methodology.

Abstract

"Comparing long-term serological markers, we found a 30% (95% CI = 21-37%) reduction in seropositivity for the RFDA intervention using a difference in differences approach incorporating serological positivity and age". I would prefer if the authors stick to the under-fives here for what was reported in the result section: "while under 5's in the RFDA arm were 19% (95% CI = 4-32%) less likely to test positive than under 5s in the RFTAT arm." Table 2 does not show a difference by treatment overall.

Introduction and methods

The introduction is compact and to the point.

However, the method section is hard to read without reading additional information from the protocol paper, the trial registry, or the protocol, and it should not be like that. The flow chart is confusing, because there is follow up for a subgroup of 30-60-90 days according to the trial registry, and we have to assume these all complied? It is also not clear what the sample size is for this subgroup in each arm. It is also not clear if the eligible persons for RFDA are for the same 140 meters as for the RFTAT group. Was there supervision of the DHAP in the RFDA arm? (This information is in the protocol paper but could be added in a few words).

The second secondary outcome is not presented in this paper (PCR parasite prevalence among individuals participating at 0, 30 and 90 days following a reactive research response for a period of 24 months)?

The initial outcome was among children <5 years of age, but this was apparently changed to <15 years of age, with a referral to the protocol paper. I screened that paper but still can't find the reason why, may have missed it. The protocol also talks about <15 years of age, but I can't find a list with changes to the protocol where this is explained.

Why were samples from infants (<1 year old) not included?

I don't really understand how the DND model was used when there was just one timepoint at the end, and not the usual two surveys comparing beginning and end, and intervention. Was child age used as a proxy for time? Given that the intervention lasted two years, were other age cut-offs tried out?

Results

The reasons for the persons not receiving the allocated intervention in the RFDA arm could be added as a footnote to Figure 1. The flowchart gives the impression of a hurried job, with these instructional texts "give reasons" still in, as if it is not yet finished. If there is 0, this instructional text could be removed.

Table 1: how can there be more than one RDC response to a confirmed malaria case?

Can there be a characteristics table in the supplement for the children, so readers can eyeball if the results by arm, especially for the factors used for adjustment?

"Adjusted model included factors of wealth quintile, head of household education, household electricity access, house had open eaves or not, whether the child slept under an insecticide-treated mosquito net the previous night and for the short term antigens age in years."

What is short term antigens age in years? Is this used here as an indicator of malaria transmission in the area?

Line 12-13: Positivity for AMA-1 or MSP1-19 alone were ~11% while GLURP-R2 alone was 27% (S6 Table). Comment: Is this table S6 or table S1? In S1, instead of showing number of positive and negative results, can be shown positive (percentage) and total sample in columns? This makes it easier to verify.

I could not find the captions and legends for the graphs belonging to the main text, and there were these missing referrals (e.g. "Error! Reference source not found.,") which made it hard to puzzle the results figures together.

"although the same AE data collection system was in place". What was the AE data collection system? This was the adherence visit at day 3? Please add to methods so it makes more sense.

Discussion

In my opinion, the following section could be moved to the methods section, to pre-empt questions:

"In the first stage, we leveraged long-lasting IgG responses (> 5-7 years) into a difference-in-differences (DID) (27) comparison, that examines the difference between pre- and post-intervention measures (the differences) between intervention and control groups (the difference). For the CoRE study this means comparing how differently the two arms of the trial changed over time. Typically, a DID approach requires a pre-intervention measure, but as we applied it here, we used seropositivity in children aged 5-15 as a measure of pre-intervention exposure."

I am not sure what the two stages are referring to in the discussion: two analyses using serological outcomes were performed according to the methods section. Stage suggests that you used one model as a precursor for the next model, but as far as I understand that was not the case. However, if I understand it wrong, I would appreciate if you could improve the language so I can understand it in one go.

Supplement

It is time consuming to click through all these separate supplemental tables and figures to be able to see them. Can they be combined in one supportive PDF file? E.g., I wanted to check if there was a characteristics table of the participants in the surveys at the end of the intervention. That will also help with the figures, because now they are presented without text underneath, and it is cumbersome to find out what they represent.

Dataset

Note that with the dataset only the unadjusted analyses can be verified, not the adjusted. I could replicate the AMA-results of Table S1.
---

## [Decision Letter · Decision Letter 1]

26 Oct 2022

Reactive focal drug administration associated with decreased malaria transmission in an elimination setting: serological evidence from the cluster-randomized CoRE study

PGPH-D-21-00549R1

Dear Dr Bridges,

We are pleased to inform you that your manuscript 'Reactive focal drug administration associated with decreased malaria transmission in an elimination setting: serological evidence from the cluster-randomized CoRE study' has been provisionally accepted for publication in PLOS Global Public Health.

Best regards,

Christian Wejse, MD, PhD, Assoc.Prof

Academic Editor

Reviewer #2: The response to review comments is adequate. ⁴